# Effectiveness of Interventions to Promote Sustainable Employability: A Systematic Review

**DOI:** 10.3390/ijerph16111985

**Published:** 2019-06-04

**Authors:** Emmelie Hazelzet, Eleonora Picco, Inge Houkes, Hans Bosma, Angelique de Rijk

**Affiliations:** 1Department of Social Medicine, CAPHRI Care and Public Health Research Institute, Faculty of Health, Medicine and Life Sciences, Maastricht University, 6200 MD Maastricht, The Netherlands; inge.houkes@maastrichtuniversity.nl (I.H.); hans.bosma@maastrichtuniversity.nl (H.B.); angelique.derijk@maastrichtuniversity.nl (A.D.R.); 2Department of Psychology, University of Milano-Bicocca, 20126 Milano, Italy; e.picco1@campus.unimib.it

**Keywords:** Sustainable employability, effectiveness, interventions, core components, vitality, health, productivity, valuable work, long-term perspective, systematic review

## Abstract

*Background:* Despite growing interest in sustainable employability (SE), studies on the effectiveness of interventions aimed at employees’ SE are scarce. In this review, SE is defined by four core components: health, productivity, valuable work, and long-term perspective. The aim of this review is to summarize the effectiveness of employer-initiated SE interventions and to analyze whether their content and outcome measures addressed these SE components. *Methods:* A systematic search was performed in six databases for the period January 1997 to June 2018. The methodological quality of each included study was assessed. A customized form was used to extract data and categorize interventions according to SE components. *Results:* The initial search identified 596 articles and 7 studies were included. Methodological quality ranged from moderate to weak. All interventions addressed the components ‘health’ and ‘valuable work’. Positive effects were found for ‘valuable work’ outcomes. *Conclusions:* The quality of evidence was moderate to weak. The ‘valuable work’ component appeared essential for the effectiveness of SE interventions. Higher-quality evaluation studies are needed, as are interventions that effectively integrate all SE core components in their content.

## 1. Introduction

Maintaining employees’ sustainable employability is important for employers and employees. The labor force is aging, and current work environments are challenged by the need for flexibility, widespread digitalization, and for building sustainable organizations [1,2]. Employers are searching for different ways to stimulate employee health in a sustainable way and to build organizations consisting of vital workers [2]. A definition of sustainable employability (abbreviated SE) has been proposed by Van der Klink and colleagues [3,4]: *“Sustainable employability means that, throughout their working lives, workers can achieve tangible opportunities in the form of a set of capabilities. They also enjoy the necessary conditions that allow them to make a valuable contribution through their work, now and in the future, while safeguarding their health and welfare. This requires, on the one hand, a work context that facilitates this for them and on the other, the attitude and motivation to exploit these opportunities”* [4] *(p. 4)*.

SE interventions should thus address at least four core components: a health component (e.g., well-being, vitality, and quality of working life), a productivity component (e.g., work ability, productivity, work engagement, and work performance), a valuable work component (e.g., positive attitude, job motivation, and having the right competences for one’s work), and, considering the long-term goal of SE, a long-term perspective component (e.g., future employability of employees of all ages and long-term effects).

The ‘valuable work’ component is derived from the capability approach of Sen [4,5]. This value-driven approach highlights what is valuable for and valued by people and how these values can be achieved in someone’s life. It is not only what an individual, in this case an employee, actually does. It also concerns what an individual can do or is able to do [5,6]. It is a shared responsibility of the employee and the work context to build up and facilitate these capabilities. These are the opportunities to achieve and enable a valuable (working) life [6].

Despite a growing interest in SE, studies on the effectiveness of SE interventions to promote SE are scarce. Workplace health promotion interventions (WHPIs) have more often been developed and evaluated, but these mainly focus on lifestyle, health, and short-term effects [7,8]. Ideally, SE interventions should include all core components of SE and thus also focus on long-term effects, as this is inherent to SE [3,9].

The effectiveness of SE interventions is less often studied. A recent review by Oakman and colleagues [10] showed that moderate-quality evidence is available for the effectiveness of interventions aimed at improving employee work ability (which can be considered a proxy of SE). A small but significant and positive effect was found, but the authors concluded that further high-quality research is needed [10]. Another review by Cloostermans [11] showed that, among aging employees, there is insufficient evidence for the effectiveness of SE interventions. What might also be lacking is a focus on SE for employees of all work ages. Prevention of diseases, having a focus on lifetime employability, and a personal career development should start at an early age [12,13].

In the current study, we aim to review the evidence on the effectiveness of employer-initiated SE interventions. This includes the analysis of the interventions’ content and the outcome measures used to evaluate their effectiveness. To what extent the four SE core components are covered in both the intervention content and in the outcome measures is specifically assessed. Each study’s methodological quality is evaluated by a multi-design quality assessment tool.

## 2. Materials and Methods

### 2.1. Search Strategy and Study Selection

Six electronic databases were searched (Cinahl (Ebsco), EconLit (Ebsco), Embase, PsycInfo (Ebsco), Pubmed, and Web of Science). The search was limited to full-text scientific articles published between January 1997 and June 2018. This time period appears to be sufficiently broad as attention for and research into SE research is relatively recent. The following keywords were used: ‘sustainable employability’ OR ‘sustained employability’ OR ‘sustainable employment’ OR ‘sustained employment’ OR ‘sustainable work’ OR ‘sustained work’. We searched for studies covering these terms in the title, abstract, or text body. When we added the keywords ‘evaluation’ or ‘intervention’ to the search (with the search command AND), we did not find enough relevant articles. We included only studies which quantitatively evaluated the effectiveness of employer-initiated SE interventions among currently active employees (whether temporarily on sick leave or not). Therefore, we did not include qualitative studies or process evaluations, although the latter were used to describe the context of the studies and interpret the absence or presence of effectiveness. Generally, to optimize the sensitivity of our search, we ensured—also in the manual selection—that our search strategy and selection was broad. Based on a screening of titles and abstracts, the initial selection of studies was done independently by the first two authors. When decisions about inclusion differed between the two authors, they met to achieve consensus about study inclusion. In case of persisting disagreement, consensus was achieved in discussion meetings with all authors, using the full text articles.

### 2.2. Methodological Quality Assessment

The methodological quality of the included studies was assessed by means of the Quality Assessment Tool for Quantitative Studies developed by the Effective Public Health Practice Project (EPHPP) [14,15,16]. This tool allows the assessment of the methodological quality of both randomized and non-randomized studies. It is suitable for use in a systematic literature review and has previously been used in other studies [15,16]. The tool consists of six criteria: selection bias at baseline, study design, confounders, blinding, data collection methods, and withdrawals and dropouts. Every criterion was assessed as “strong”, “moderate”, or “weak”. The appropriateness of the statistical analyses was assessed separately: “yes” or “no”. As per the EPHPP protocol, the overall quality rating was determined by assessing all criteria ratings, except the data analysis. A study with at least four strong ratings and no weak ratings was assessed as “strong”; a study with less than four strong ratings and one weak rating was assessed as “moderate”; and a study with two or more weak ratings was assessed as “weak”. Two first two authors independently rated the studies. The results were compared and differences were discussed during a consensus meeting. The three last authors additionally assessed three, two, and two articles, respectively, and their results were compared to those of the two first two authors. In order to reach consensus, differences were discussed with all authors. Hence, all studies were assessed by three reviewers.

### 2.3. Data Extraction

Using a customized form, the first author extracted the data from the studies. The form included the following captions: target population (N and sub-populations), follow-up period, the content of the interventions, the outcome measures that were used, and the effectiveness of the interventions. We categorized each study according to which of the four SE core components were covered in the content of the intervention and in the set of outcome variables used to evaluate the effectiveness. Table 1 shows an operationalization of the four SE core components.

## 3. Results

### 3.1. Selection of Articles

A total of 596 records were retrieved. After removing 224 duplicates, 372 unique references remained. Based on title and abstract, 25 articles were selected for potential inclusion (Figure 1). Of these 25 articles, 18 were excluded because they did not report an intervention (7 articles), the intervention was not a SE intervention (3 articles), the intervention was not evaluated on effectiveness (5 articles), the population did not meet our inclusion criteria (2 articles), or the articles were not scientific articles (1 article). In total, seven articles were included in this review. We also screened the reference lists of these seven articles. This search did not result in additional articles. See Figure 1 for the Systematic Reviews and Meta-Analysis (PRISMA) flow diagram.

### 3.2. Methodological Quality of the Studies

In general, the overall methodological quality of the seven studies ranged from moderate to weak (Table 2). Three out of seven studies [17,18,19] had a moderate overall methodological quality. One study scored “strong” four times on the criteria [18]. However, blinding of participants and researchers was not possible in any of the studies. This was rated as weak which at best leads to an overall moderate study quality. Four studies [20,21,22,23] had a weak overall methodological quality which was mainly due to selection bias, no blinding of participants or outcome assessors, and a low follow-up rate. One of these four studies was very weak, scoring low on five out of six criteria [23]. This was due to the lack of a description of the tool properties, confounders, dropout rates, and data analysis techniques. The remaining six studies used appropriate data analysis techniques. Upon request, more information about the rating of each criterion is available from the authors.

### 3.3. Data Extraction

Table 3 provides an overview of the general characteristics of the interventions, their content, the outcome measures of the evaluation, and the interventions’ effectiveness.

### 3.4. Content and Effectiveness of SE Interventions

The interventions varied and included both individual and workplace interventions. To support SE of employees in the construction sector, Oude Hengel et al. [17,18] evaluated a worksite prevention program to improve work ability and health-related quality of life. The intervention consisted of three components: two physical components and one mental component. The employee received two individual training sessions with a physical therapist to lower the employee’s physical workload and the sessions included a quick scan and a job observation at the workplace. Afterwards, advice was given to the employee. In the second training, the experiences of the employees were discussed. The second physical component was a rest break tool to improve the employee’s ability to balance between work and recovery. The mental intervention component consisted of two empowerment group training sessions to increase the employee’s influence at the worksite [17,18]. In the first training session, employees wrote down a list of topics that they thought were amenable to change and they agreed on an action plan. In the second training, the action plan was evaluated. Overall, this intervention showed no effect on work ability, health, work engagement, social support, and need for recovery. A negative effect was found for the physical workload after 6 months of follow-up.

The study of Koolhaas et al. [19] evaluated a problem-solving based intervention focused on enhancing the capacity and awareness towards SE of aging employees. First, an inventory of work-related problems and a needs assessment was performed. Afterwards, a dialogue between the employee and the supervisor was performed to discuss solutions followed by an action plan. For the preparation of the dialogue and development of the action plan, a booklet was provided to the employees. The supervisors were trained to challenge the workers to reflect on the feasibility of solutions. Furthermore, knowledge on SE and problem-solving techniques were discussed with the supervisors. The problem-solving based intervention showed a positive effect on the secondary outcome measures of perceived work attitude, skill discretion, and self-efficacy, whereas no effect was found on the primary outcome of productivity and a negative effect was found for work ability and vitality [19].

Van Holland et al. [20] evaluated an intervention program to identify employees who are at risk for reduced SE. The program consisted of different screening tests, such as a digital questionnaire on work ability, health and lifestyle, and physical measurements, such as biometric and functional capacity measures. In a counselling session with a vocational physiotherapist, the results of the screening tests were discussed with the employee and, when necessary, the employee received advice on whether or not to take consecutive actions. The intervention program showed negative effects on sickness absence and productivity, whereas a small positive effect was shown on the psychosocial outcome meaning of work, a measurement component of psychosocial workload [20].

A longitudinal study by Van der Meer and colleagues [21] evaluated the impact of SE company policies on work engagement and word ability of aging employees. These policies were especially designed for aging employees to support their SE. Employees received an online questionnaire about different topics, such as health and productivity. Furthermore, employees were asked whether the two SE company policies for aging employees: (1) “reduced number of working hours per week” and (2) “exemption from evening or night work”, were available and used in their company. The SE company policy “exemption from evening or night work” resulted in a structural change and a higher work engagement after one year. However, the SE policy “reduced number of working hours per week” showed a negative effect on work ability among older employees [21].

The study of Van Scheppingen et al. [22] evaluated a large-scale intervention to induce a health- promoting organizational change process in a population of employees in a dairy company. The intervention consisted of three main components: (1) dialogue sessions aimed at reflecting on the value of health and vitality at work among employees and at putting this on the personal agenda of employees and the organization, (2) vitality-promoting activities at the department level, such as lunch walks or workshops on healthy work postures, and (3) physical activities in which employees could participate individually, such as running races and team sports activities. The different intervention components showed positive effects on the outcomes openness toward health, smoking, healthy eating, bonding social capital, and perceived sustainable employability [22].

Lastly, using an online questionnaire filled out by employees, a longitudinal study by Weiss [23] evaluated the progress of companies regarding health, safety, sustainability, and stewardship. Monthly best practice exchange meetings between companies were organized to promote collective efficacy by sharing ideas about the four areas. This study shows that a collective efficacy approach seems to improve the health and sustainable work culture and to increase employee attachment to the organization [23].

### 3.5. Content and Effectiveness of SE Interventions in the Light of the Four SE Core Components

This section describes patterns in the effectiveness of the SE interventions, taking into account the methodological quality of the studies and the extent to which the four SE core components are covered in both the intervention content and the outcome measures.

In the three studies of moderate quality (2 interventions), both interventions included the following three SE components: ‘health’, ‘valuable work’, and ‘long-term perspective’. All four components were measured as outcomes. Two studies showed significant negative effects on the ‘health’ outcomes (1 of 2 measures and 1 of 2 measures, respectively) [17,19]. One study showed a significant negative effect on the ‘productivity’ outcomes (1 of 3 measures) [19]. The latter study showed positive significant effects for all measures of ‘valuable work’ outcomes, though [19]. All three studies used a follow-up period of one year with repeated measurement points.

Of the four weak studies (4 interventions), all interventions included the ‘health’ and ‘valuable work’ component [20,21,22,23] and two interventions included the ‘productivity’ component along with the ‘health’ and ‘valuable work’ component [20,21]. The ‘long-term perspective’ component was covered in one intervention, as the intervention included all working ages [22]. Overall, all four SE components were measured. In the weak studies, one study showed a significant positive effect on ‘health’ outcomes (2 of 2 measures), ‘valuable work’ outcomes (3 of 3 measures), and ‘long-term perspective’ outcome (1 of 1 measure) [22]. It also reported how specific intervention ingredients were related to effectiveness [22]. One study showed a significant negative effect on ‘health’ outcomes (1 of 3 measures) and on ‘productivity’ outcomes (1 of 2 measures), while a small significant positive effect was shown on the ‘valuable work’ outcome (1 of 1 measure) [20]. One study showed a significant positive effect as well as a negative effect on the two ‘productivity’ outcomes [21].

The content of all interventions addressed the components ‘health’ and ‘valuable work’. Regarding ‘long-term perspective’, two interventions included employees of all work ages, and three interventions included employees of 45 years and older. Positive effects were found for ‘valuable work’ outcomes and, to a lesser extent, for ‘health’ outcomes and ‘productivity’ outcomes. Also negative effects were shown for ‘health’ and ‘productivity’ outcomes. Regarding the ‘productivity’ outcomes, the chosen outcome measures were not always in line with the intervention content. More precisely, the ‘productivity’ component was absent in the majority of interventions (4 interventions). The studies that included three SE core components in the content of the SE interventions led to fewer effective outcomes (not even with a longer follow-up period) compared to the studies that included only two SE core components.

## 4. Discussion

This literature review systematically summarizes available evidence regarding the effectiveness of employer-initiated SE interventions. First, we analyzed the content and effectiveness of the SE interventions. Second, we analyzed the extent to which these interventions covered the four SE core components in their content, and whether these components were addressed in the outcome measures used to evaluate effectiveness. A relatively low number of studies are available that evaluated SE interventions and our findings indicate a moderate to weak quality of evidence on the overall effectiveness of SE interventions. Mixed effects were found, in which the majority of the studies showed negative or no effects on ‘health’ and ‘productivity’ outcomes. A minority showed significant positive effects, which were mainly interventions having a ‘valuable work’ component in their content and outcome measures. The limited effectiveness is in line with earlier research (in aging employees [12]). There might be several causes for the limited effectiveness, related to the content of the SE interventions, program failure, and choice and operationalization of outcome measures.

Firstly, based on the definition of SE by Van der Klink and colleagues, we distinguished four SE core components (i.e., health, productivity, valuable work, and long-term perspective). At least two SE core components, ‘health’ and ‘valuable work’, were addressed in the content of all interventions. Regarding the effectiveness and potentially effective ingredients of the moderate-quality studies, the study of Koolhaas et al. [19] for instance showed positive effects on ‘valuable work’ outcomes, which might be due to ‘valuable work’ components in the intervention content. Specifically, the first two intervention ingredients, the inventory of problems and the dialogue between employee-supervisor, might have been be potentially effective ingredients, as—in terms of awareness and own responsibility for SE—the intervention changed the employees’ perspective positively. However, the study showed negative effects on ‘health’ as well. In general, the negative effects of an intervention might be explained by a response shift of the employees who, as extra attention is being paid to health, become more aware and responsible for their health and related problems. Another reason for the negative effects on ‘health’ might be due to the short follow-up [24]. To assess the full effect on health, long-term studies (e.g., decades) would be needed. Regarding the study of Oude Hengel et al. [17,18], a reason for the lack of effect of the intervention may be due to a healthy worker effect, as the health and work ability of the employees at baseline were considered good [18]. The weak studies showed positive effects on ‘valuable work’ outcomes as well, and one study revealed which specific intervention ingredient contributed to which positive effect. Valuable work components, such as the dialogue and reflective thinking sessions, appeared effective ingredients in this intervention study [22]. In line with the value-driven approach of Sen, ‘valuable work’ appears to be addressed effectively in three intervention studies. It seems that SE interventions including ‘valuable work’ enable a valuable work life and are as such appreciated by employees. All ‘valuable work’ outcomes are related to the individual level. However, it might well be that the work context rather than the individual facilitated these outcomes via the ‘valuable work’ component in the intervention. It is important to include a ‘valuable work’ component in SE interventions.

Van Holland et al. [20] reflected on the negative effects of their intervention on ‘health’ and ‘productivity’. The negative effect on ‘health’ outcomes (sickness absence) might be explained by the investments (aimed at reducing sickness absence) that the participating company already did prior to the intervention. There might not have been room for additional improvement. In the study evaluating SE company policies [21], both a positive and a negative effect were found on the ‘productivity’ outcomes. One policy decreased the work ability in older employees. The authors speculated that this might be due to the fact that the policy was not tailored to the needs of these employees. It might also be that employees who started to work less hours, felt less productive as a result. In addition, the authors reported that aging employees who were eligible for the policy may have perceived feelings of being “superfluous”. The SE policy “exemption from evening/night work” showed a positive effect on work engagement. This is may be due to employees feeling more energized after quitting evening/night shifts [21].

Secondly, process evaluations may provide more insight into the facilitators and barriers in SE interventions or difficulties in the implementation process [25,26]. We considered whether authors performed such process evaluations and/or otherwise provided explanations in their reports for the lack of effectiveness. In our review, four out of seven studies [17,18,19,20] included a process evaluation, and the authors mentioned a variety of possible program failures, such as a poor implementation of a specific intervention content, low compliance, or whether the intervention is delivered as intended [17,20]. The influence of contextual factors appears to play a role as well, for example in the intervention of Oude Hengel et al. [17,18], where an economic recession negatively influenced the dose received of the study. However, a smaller company size or higher management engagement led to higher attendance rates [27]. The ineffectiveness in the study of Van Holland may be explained by a poor follow-up of recommendations of the participants [20,28]. Negative effects on vitality and work ability as primary outcomes were explained by a low adherence of the workers in the last step of the intervention [19]. Additionally, dose-delivered issues occurred, such as a training duration which was too short and the level of skills and knowledge of supervisors, which might have been inadequate [19].

Finally, the limited effectiveness of interventions might also be explained by the fact that the choice and operationalization of outcome measures did not align with the intervention content. This is particularly salient for the ‘productivity’ component which often is absent in intervention content and outcome measure.

### 4.1. Study Strengths and Limitations

One methodological strength is the systematic search of the literature. In this review, not only RCTs but also alternative study designs such as a quasi-experimental or cohort studies were included [10,26,29]. Although a RCT is the golden standard to determine intervention effectiveness, in the field of organizational work site interventions, it could act as a limiting factor, and alternative designs often deployed as randomization, controlling, and blinding are often difficult or even impossible. When using the EPHPP, the highest methodological quality that can be achieved in this field of research is moderate as a consequence. This should be taken into account when judging our evaluation of the quality of the studies included. Without the blinding criterion, one study would have scored a strong overall quality [18].

Furthermore, the number of SE interventions and evaluation studies is still very low. We could only include seven studies that matched our inclusion criteria. Our search might have been too narrow or the manual selection could have limited the number of hits unnecessarily. During the manual selection, studies outside the research field showed up (e.g., studies focused on sustainability in terms of improvement of the planet/environment). Therefore, we explicitly focused on the combination of the terms sustainability/sustainable and work/employment in the selection of the papers. Furthermore, as we were also interested in the SE intervention content and whether that showed any relation to effectiveness, we rather broadly and thus sensitively included all interventions that were explicitly framed as SE and focused on the level of employees. As we did not focus just on employees in a specific target population in a sector or an age group, we further increased this sensitivity. The disadvantage of this broad selection with still few hits is that the SE interventions are diverse, which complicates the detection of patterns of effectiveness. Although we were explicit about how we registered the SE core components in both intervention content and outcome measures, we acknowledge that the reliability and validity can still be questioned. The systematic manner in which we addressed the four suggested SE core components, both in intervention content and in the outcome, and in which we also assessed the link between intervention and outcome measures for potentially effective ingredients, is a clear strength of this study. This is the first study applying this method systematically, as far as we know. Moreover, this is the first time that the definition of SE was further operationalized into four SE core components in relation to SE interventions. This way of operationalization seems to be most in line with the SE definition of Van der Klink [6]. There is no consensus yet among scholars on how to operationalize SE though.

Furthermore, no distinction could be made between subgroups (i.e., educational level). Therefore, it is not possible to make statements about the differential effectiveness in specific subgroups, which could have been relevant as specific SE interventions or ingredients might be more effective for specific subgroups.

Finally, most of the included intervention studies were conducted in the Netherlands. One explanation might be that the concept of ‘employability’ was introduced in the public debate already in the 1990s in the Netherlands [30]. At that time, employees were thought to invest in their own employability during their whole working career, due to societal and legislative developments and a government withdrawing from labor-related issue [30]. Further, employers in the Netherlands traditionally have a large responsibility for the health and age management in the workplace. In the last decade, the concept of SE has been embraced by employers because of these developments as a solution for working population becoming older and—if not prevented—less productive. Research has followed these interests.

### 4.2. Recommendations for Future Research and Practical Implications

This review has several implications for future research and practice. It appears to be difficult to perform high-quality research in this field. Researchers should pay attention to designing studies with the highest quality possible, given the circumstances. Designing a RCT might not be possible, but other methodological criteria should be met as good as possible. For example, participation rates should be as high as possible, to minimize the selection bias. A high follow-up rate is important as well. However, organizational changes in a work setting could affect the follow-up rate and the potential effectiveness of the intervention [31]. As blinding is difficult in a workplace setting, researchers could minimize the problems related to this by providing no information about the main research question to the study population. Furthermore, this review focused especially on SE interventions at employee level. It might be interesting to look at SE interventions at other levels of organizations, for example at the level of the managers [32].

In the SE studies in this review, including more SE core components in the content of the interventions was not related to more effective outcomes. This might be due to the choice and measurement of the outcomes, and the inconsistent aligning of the intervention content and outcome measures. Future SE interventions should be developed which preferably better integrate the SE core components and address them in the outcome measures as well, to frame well-considered SE interventions and evaluations.

A full-process evaluation should be an integral part of a SE intervention, to explain both the (lack of) effectiveness and to understand the implementation process in terms of possible program failure. Further research might focus on whether more comprehensive SE interventions (i.e., including all SE core components) are more effective or whether specific intervention ingredients are more effective. Research should not only focus on employees with fixed contracts; it also needs to examine specific (precarious) occupational groups (e.g., younger employees, self-employed, or employees with a flexible contract) as these populations are growing.

Defining and conceptualizing SE is ongoing. Both in the SE definition and in the interventions studied, ‘valuable work’ appears to be effective. However, the longitudinal and long-term nature of SE in particular receives little attention. Many SE researchers have only addressed older employees, as our review confirmed (except one study). In accordance with the SE definition, we think that SE interventions cannot start early enough, definitely prior to the occurrence of chronic diseases that are prevalent in older workers [33]. Employers are advised to focus on SE and prevention as early as possible in an employee’s career, as it will be beneficial to improve the employability of an employee later in life [34]. Longer follow-up periods (in both intervention and research) are highly recommended. In particular, effects on health might be the result of a long-lasting process. The currently more flexible and dynamic labor market might be a practical factor hindering long-term follow-ups for many employees. Researchers might consider the use of online surveys via national tax or social security registers to perform longitudinal intervention research.

This review focused specifically on employer-initiated SE interventions to promote employees SE. As mentioned above, the responsibility for SE is shifting, and different stakeholders with different interests are involved (employee; employer; government). Employers and employees could have a shared responsibility to improve SE, in which employees take their own responsibility, and the employers should enable a supportive work context to do so [4]. In a dynamic environment, taking care of employees’ SE might not be the sole responsibility of employers anymore. The government and social partners should also play a role in terms of SE policy development [35]. SE will increasingly become a joint effort of multiple stakeholders. Employers and governments could play a role to address early employability awareness among younger employees. The self-employed employees might also be of interest for the government, for whom it would create awareness, provide campaigns, and develop regulations. All people of working age should become more aware as this could be beneficial for their later working career and might influence long-term improvements [36].

## 5. Conclusions

Employers, employees, and social partners are facing a challenging and dynamic labor market in which SE is becoming increasingly important. Employers develop or buy and implement SE interventions to improve employees’ SE. This review found only moderate to weak evidence for the effectiveness of employer-initiated SE interventions. The number of SE interventions is limited, and most do not incorporate all four core components of SE (i.e., health, productivity, valuable work, and long-term perspective). Positive effects were shown on the ‘valuable work’ outcomes. More attention is needed on the development of higher quality SE interventions and building a more solid evidence base for the effectiveness of those interventions, which might be beneficial for stimulating employees’ SE.

## Figures and Tables

**Figure 1 ijerph-16-01985-f001:**
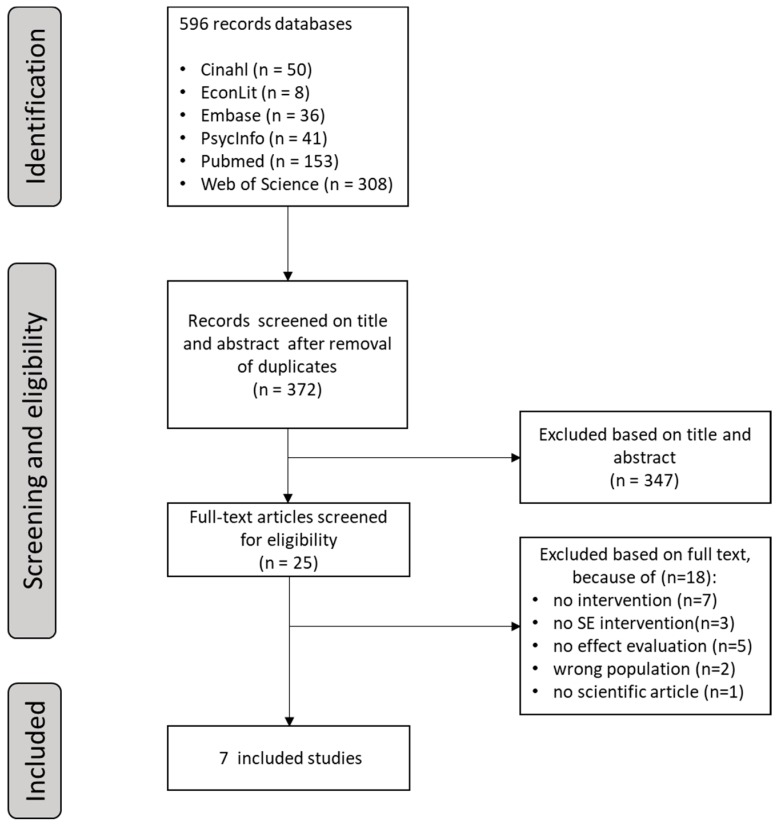
Selection of studies: Systematic Reviews and Meta-Analysis (PRISMA) flowchart.

**Table 1 ijerph-16-01985-t001:** Operationalization of sustainable employability (SE) core components in intervention content and outcome measures.

SE Core Component	Intervention Content	Outcome Measures
Health	Intervention focuses on health aspects, such as well-being, quality of working life, vitality, lifestyle, or mental and physical health.	E.g., well-being, quality of working life, vitality, lifestyle, or mental and physical health.
Productivity	Intervention focuses on productivity aspects, such as work ability, productivity, or work engagement.	E.g., work ability, productivity, or work engagement.
Valuable work	Intervention focuses on valuable work aspects, such as perceived positive attitude, job motivation, having the right competences to perform the job, and development of skills and knowledge.	E.g., perceived positive attitude, job motivation, having the right competences to perform the job, and development of skills and knowledge.
Long-term perspective	Intervention focuses on all work ages. Intervention explicitly aimed at long-term effects.	Use of a follow-up period (at least 1 year) with repeated measures not only assessing short-term effects.

**Table 2 ijerph-16-01985-t002:** Methodological quality of the studies included.

Study	Selection Bias (Baseline)	Study Design	Confounders ^b^	Blinding	Data Collection	Withdrawals and Dropout	Data Analysis	Overall Quality ^c^
Oude Hengel ^a^ [17]	Strong	Strong	Strong	Weak	Moderate	Moderate	Yes	Moderate
Oude Hengel ^a^ [18]	Strong	Strong	Strong	Weak	Strong	Moderate	Yes	Moderate
Koolhaas [19]	Moderate	Moderate	Strong	Weak	Strong	Moderate	Yes	Moderate
Van Holland [20]	Weak	Moderate	Strong	Weak	Strong	Weak	Yes	Weak
Van der Meer [21]	Weak	Moderate	Strong	Moderate	Moderate	Weak	Yes	Weak
Van Scheppingen [22]	Weak	Moderate	Strong	Weak	Strong	Weak	Yes	Weak
Weiss [23]	Weak	Moderate	Weak	Weak	Weak	Weak	No	Weak

^a^ Same intervention, but different outcome measures; **^b^** Were demographics and pre-intervention outcome scores taken into account as confounders?; ^c^ Overall quality: Strong (4 strong and no weak ratings); Moderate (<4 strong ratings and one weak rating); Weak (two or more weak ratings).

**Table 3 ijerph-16-01985-t003:** Description of interventions, outcome measures and effectiveness.

Study	Study Population	Follow-Up	Intervention Content	*SE Core Components in Content*	Outcome Measures	*SE Core Components in Outcome Measures*	Effectiveness ^b^
Oude Hengel ^a^ [17] Moderate overall quality	Construction workers (*N* = 293) Mean age = 41.8 years intervention group and 44.2 years control group Education level: Intervention group: Low (74%); Medium-high (26%). Control group: Low (84%); Medium-high (15%).	3,6,12 months	Two individual training sessions with a physical therapist to lower physical workload ^c^. Training 1 - Health risk assessment (quick observation scan) - Individual advice and max. 3 recommendations Training 2 after 4 months - Discuss experience and impact of former advice.	*Health*	Physical workload	*Health*	**Negative effect** (in intervention group 6 months of follow-up
					Need for recovery	*Health*	No effect
					Work engagement	*Productivity*	No effect
					Social support at work	*Valuable work*	No effect
			2. A rest-break tool on fatigue and need for recovery. Four steps: - Workers’ own expectations about their fatigue - Short-term advice to take mini-rest breaks - Selection of possible causes of fatigue - Long-term advice about structurally lowering fatigue.	*Health* *Long-term perspective*			
			3. Two empowerment training sessions to increase worker’s influence at the worksite. Five steps: - Introduction of self-efficacy. - Introduction of the training. - Explanation of how to change passive attitude to pro-active and positive attitude. - List of topics workers would like to change during the intervention - Action plan	*Valuable work*			
Oude Hengel ^a^ [18] Moderate overall quality	Construction workers (*N* = 293) Mean age = 41.8 years intervention group and 44.2 years control group Education level: Intervention group: Low (74%); Medium-high (26%). Control group: Low (84%); Medium-high (15%).	3,6,12 months	Two individual training sessions with a physical therapist to lower physical workload ^d^.	*Health*	Sick leave	*Health*	No effect
					Musculoskeletal symptoms	*Health*	No effect
			2. A rest-break tool on fatigue and need for recovery ^d^.	*Health* *Long-term perspective*	Mental and physical health status	*Health*	No effect
					Work ability	*Productivity*	No effect
			3. Two empowerment training sessions to increase worker’s influence at the work site ^d^.	*Valuable work*			
Koolhaas [19] Moderate overall quality	Aging workers (Age >45 years) (*N* = 125) Education level: Low (17%) Medium (40%) High (43%)	1 year	Inventory of work-related problems, needs and career and personal development opportunities of the worker.	*Health* *Valuable work*	Perceived fatigue	*Health*	No effect
					Vitality	*Health*	**Negative effect**
			2. Dialogue between worker and supervisor to discuss solutions; Supervisors were trained in challenging the workers to reflect on the feasibility of solutions.	*Valuable work*	Work ability	*Productivity*	**Negative effect**
					Productivity	*Productivity*	No effect
					Work engagement	*Productivity*	No effect
			3. Making an action plan to plan and implement solutions for a follow-up period next year.	*Long term perspective*	Job content (skills discretion)	*Valuable work*	**Positive effect**
					Perceived work attitude	*Valuable work*	**Positive effect**
					Self-efficacy	*Valuable work*	**Positive effect**
Van Holland [20] Weak overall quality	Workers of Dutch meat processing company (*N* = 305) mean age = 50.6 years Education level: No-low (64%); Medium-high (32%)	3 years	Risk assessment tests to create the risk profile of the employee, such as: - Tests on physical and mental health (biometric measures) - Tests on physical and mental work capacity (functional capacity) - Assessment on work ability, health and lifestyle.	*Health* *Productivity*	Sickness absence	*Health*	**Negative effect**
					Health	*Health*	No effect
					Vitality	*Health*	No effect
			2. Counselling session. The employee receives feedback on his/her results from the screening tests by a consultant and advice on whether or not to take consecutive actions.	*Valuable work*	Work ability	*Productivity*	Negative effect
					Productivity	*Productivity*	**Negative effect**
					Psychosocial variable: meaning of work	*Valuable work*	**Positive effect**
Van der Meer [21] Weak overall quality	Workers, including self-employed and people without paid job (45–64 years) (*N* = 6922) Mean age = 53.7 years	2 years	Create awareness and knowledge of aging employees on the availability and the use of two company policies to support: - ‘reduced working hours per week for older workers’ - ‘exemption from evening or night work for older workers’.	*Health* *Valuable work* *Productivity*	Work engagement	*Productivity*	**Positive effect** **(by starting to use the policy ‘exemption from evening/night work’)**
					Work ability	*Productivity*	**Negative effect:** **(by starting to use the policy ‘reduced working hours’)**
Van Scheppingen [22] Weak overall quality	Workers in Dutch dairy company (*N* = 324) Age: <30 = 14.8% 30–45 = 37.3% >45 = 47.8% Educational level: Primary = 22.2% Secondary = 42.0% Higher = 35.8%	18 months	Dialogue and reflective thinking on the value of health and vitality at work;	*Health* *Valuable work*	An improvement of employees’ lifestyle: - Physical activity - Smoking - Alcohol use - Healthy eating - Relaxation	*Health*	**Positive effect (smoking and healthy eating; component 1)**
			2. Collective vitality-promoting activities at department level;	*Health*			**Positive effect (healthy eating; component 2)**
			3. Physical activities organized at organizational level (participation on an individual basis).	*Health*	Health and vitality at work: - Perceived health - Emotional exhaustion - Vitality at work - Sustainable employability	*Health* *Long-term perspective*	**Positive effect (sustainable employability; component 1)**
					Autonomous motivation toward a healthy lifestyle	*Valuable work*	**Positive effect (component 3)**
					Bonding social capital	*Valuable work*	**Positive effect (component 1)**
					Openness toward health and vitality at work	*Valuable work*	**Positive effect (component 1)**
							**Positive effect (component 3)**
Weiss [23] Weak overall quality	Several companies with 100 or more employees No further information on demographics	4 years	The Attach21 survey tool to indicate the status quo situation on four areas: health, safety, sustainability and stewardship. Monthly best practice exchange between companies to promote collective efficacy by sharing ideas about health, safety and sustainability best practices.	*Health* *Valuable work* *Health* *Valuable work*	Six core components: Consistency Stability Confidence Trust (self-efficacy) Dedication Attachment	*Valuable work* *Long-term perspective*	No effect: high level of self-efficacy No effect: High levels of attachment

^a^ Same intervention, but different outcome measures; ^b^ Bold means statistically significant. No statistically significant results are listed as no effect; ^c^ Information obtained in other design paper. ^d^ Same intervention content in article Oude Hengel 2012.

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
