# Peer review of "Effectiveness of Interventions to Promote Sustainable Employability: A Systematic Review"

_ijerph, 2019, doi:10.3390/ijerph16111985_

Round 1

Reviewer 1 Report

This is an interesting systematic review of studies on the effectiveness of interventions to promote sustainable employability. I have a few comments and suggestions to adjust the manuscript, but otherwise I think this manuscript is ready for publication.

I would advise the authors to describe the seven studies in their review in more detail. The paragraph “Content and effectiveness of SE interventions” is quite abstract. I would like to see a more comprehensible description of the content of the interventions and their results in this paragraph.

I believe a strong point of the review is the systematic nature in which the EPHPP protocol is followed. However, a drawback of this approach is that the RCT is taken as a golden standard, which is not really suitable for organizational interventions. Moreover, important additional criteria, such as a process evaluation, quantification of the level of exposure to the intervention, and identification of active ingredients are not taken into account in this protocol. The authors are well aware of these limitations, and adequately discuss these issues in the Discussion. However, I would advise them to raise these issues in the Introduction, and to give an explanation of why the chose the EPHPP protocol despite these limitations.

Detailed comment: In Table 3, the intervention of van der Meer [21] is described as a questionnaire. This seems a bit strange. I would only describe the questions on the interventions in the Table, and describe the design of this study in the text.

Author Response

Dear Reviewer,

RE: [IJERPH] Manuscript ID: ijerph-479016 - Minor Revisions

Thank you for the detailed and constructive feedback on our paper. We think that taking account of the editorial and reviewers’ feedback has resulted in an improved paper. Below you will find the original comments followed by our responses of both reviewers (in red font), including the report of where in the manuscript we edited the text (page and line numbers) visible by Track changes. After revision, the English language (US) of the revised manuscript is critically reviewed again (highlighted as yellow in the revised manuscript).

With kind regards,

Emmelie Hazelzet,

also on behalf of the co-authors

Response to Reviewer 1:

Point 1: I would advise the authors to describe the seven studies in their review in more detail. The paragraph “Content and effectiveness of SE interventions” is quite abstract. I would like to see a more comprehensible description of the content of the interventions and their results in this paragraph.

·         Author response 1: We agree with the reviewer that the result section is quite abstract. In the result section we added an extra paragraph which describes a more detailed description of the content of the intervention and their results/outcomes. Changes to the manuscript: Yes, p. 10 line 162- 218 (result section).

Point 2: I believe a strong point of the review is the systematic nature in which the EPHPP protocol is followed. However, a drawback of this approach is that the RCT is taken as a golden standard, which is not really suitable for organizational interventions. Moreover, important additional criteria, such as a process evaluation, quantification of the level of exposure to the intervention, and identification of active ingredients are not taken into account in this protocol. The authors are well aware of these limitations, and adequately discuss these issues in the Discussion. However, I would advise them to raise these issues in the Introduction, and to give an explanation of why the chose the EPHPP protocol despite these limitations.

·         Author response 2: We have added the following at the end of the Introduction “Using a multi-design quality assessment tool, we also set out to evaluate each study’s methodological quality.”, as we think the EPHPP is an assessment tool aimed at evaluating multiple designs and has also previously been used for similar purposes. Changes to the manuscript: Yes, p. 2 line 69-70.

·         We agree with the reviewer that its content and the criteria are still dominated by a view in which RCTs are considered superior. We already addressed this limitation in the Discussion. See p. 13, line 328-336. Changes to the manuscript: No.

·         Process evaluations are indeed not part of the criteria in the EPHPP assessment scheme, but we now have much more clearly indicated that we used information of related process evaluations in reflecting upon the findings, including the contextual factors that might be of importance. Changes to the manuscript: Yes, p. 12  line 307-322. [Discussion]

Point 3: Detailed comment: In Table 3, the intervention of van der Meer [21] is described as a questionnaire. This seems a bit strange. I would only describe the questions on the interventions in the Table, and describe the design of this study in the text.

·         Author response 3: We agree with reviewer. The intervention study of van der Meer is focusing on company policies to support sustainable employability in older workers. They investigated whether these kind of policies in a company contributed to higher work ability and work engagement in older workers. They examined the use and availability of these policies among ageing employees in Dutch companies. We addressed more clearly the intervention in the table and explained the study in the text. Changes to the manuscript: Yes, table 3 p.8 and description study p. 10 line 195-203.

Reviewer 2 Report

Dear authors,

I read your article on effectiveness of interventions to promote sustainable employability. I think it is worthwhile reading and a relevant topic. The core of the paper relates to 7 intervention studies. Strikingly, it seems that a large part of these are conducted in the Netherlands. Can you give an explanation for that? In my view, also in Scandinavian countries a lot of studies have been conducted. One of these is by Dellve, L. and Eriksson, A. (2017). [ Health-Promoting Managerial Work: A Theoretical Framework for a Leadership Program that Supports Knowledge and Capability to Craft Sustainable Work Practices in Daily Practice and During Organizational Change. Societies, 7, 12. doi:10.3390/soc7020012]

Can you explain why most of the studies seem to be conducted in the Netherlands. Is sustainability especially topical in that country?

Your paper starts out by presenting the definition of sustainable employability by Van der Klink et al.. Their definition results from the capability approach by Sen which can be viewed a value-driven approach to work. In the discussion part of your paper, this approach is not much touched upon. Could you elaborate more on this lense and reflect on the results of the study in light of this lense?

Your key words of searching for research that could fit your goal did not include sustainability, or sustainable HRM. I figure that you might have looked at all titles including 'sustain#' but this is not mentioned. Could you have missed some reference (also in view of the fact that you only found 7 and that these were primarily Dutch')?

Why did you only focus on quantiative studies?

I was wondering whether you can explain the 'operationalisation' of your key concepts. I know they are derived from the definion by Van der Klink et al. but I feel that some indicators are not in the right box. For example, quality of working life could also relate to valuable work. The right competences could also be in the box of productivity. Moreover, in some writings, vitality is a separate dimension of sustainable work, reflecting work engagement and motivation.  So, is this the right conceptualisation and the rigth operationalisation of the four core concepts? May be you could elaborate on this.

Table 3 covers several pages and reading the tables is difficult when the heading of the column is presented some pages before. I wonder whether it would be better to present the 7 studies each in a separate table, since it is hard to have an overview.

Moreover, in the table it says training session, but the context information is missing. So may be the content of the intervention could be discussed in more detail, either in the text or in the table.

p. 10, line 166: long-term perspective seems to be equaled to including all ages. I am not sure whether that makes sense.

p. 10, line 182 and beyond. I would like to know why you think that the intervention can have negative effects. There is little explanation on the out omes of the interventions. I think that this is valuable information that should be provided per study and with 7 that is doable. What is the underlying mechanism that is presented in each of the 7 studies or the context factors that explain the outcomes of the intervention. That would be very helpful for scholars and practioners who want to learn from these studies and from your overview and evaluation. I think some thicker information would be of added value in that regard.

p. 11, line 210: Are the contextual factors that might have disturbed the intervention mentioned in the results section. I might have missed these, but I think the outcomes of your analysis should be presented in the results section.

p. 12. At page 12 you raise a lot of interesing and important issues. However, you do not provide an answer or some hints regarding the direction that one could take. For example, what do researchers need to do when workers have temporary jobs, are freelancing et cetera when it comes to longitudinal intervention research?

p. 13 the references miss important information. Please check the reference list. 

Author Response

Dear Reviewer,

RE: [IJERPH] Manuscript ID: ijerph-479016 - Minor Revisions

Thank you for the detailed and constructive feedback on our paper. We think that taking account of the editorial and reviewers’ feedback has resulted in an improved paper. Below you will find the original comments followed by our responses of both reviewers (in red font), including the report of where in the manuscript we edited the text (page and line numbers) visible by Track changes. After revision, the English language (US) of the revised manuscript is critically reviewed again (highlighted as yellow in the revised manuscript).

With kind regards,

Emmelie Hazelzet,

also on behalf of the co-authors

Response to Reviewer 2:

Point 1: The core of the paper relates to 7 intervention studies. Strikingly, it seems that a large part of these are conducted in the Netherlands. Can you give an explanation for that? In my view, also in Scandinavian countries a lot of studies have been conducted. One of these is by Dellve, L. and Eriksson, A. (2017). [ Health-Promoting Managerial Work: A Theoretical Framework for a Leadership Program that Supports Knowledge and Capability to Craft Sustainable Work Practices in Daily Practice and During Organizational Change. Societies, 7, 12. doi:10.3390/soc7020012]

Can you explain why most of the studies seem to be conducted in the Netherlands. Is sustainability especially topical in that country?

·         Author response 1: As reported to the editor, most of the studies were indeed conducted in the Netherlands. One explanation might be that the concept of employability was introduced in the public debate already in the 90s in the Netherlands. At that time, employees were thought to invest in their own employability during their whole working career, due to societal and legislative developments and a government withdrawing from labour-related issues. Further, employers in the Netherlands have had a large responsibility for the health and age management in the workplace. In the last decade, the concept of sustainable employability has been embraced by employers because of these developments as a solution for working population becoming older and – if not prevented – less productive. Research has followed these interests.  Changes to the manuscript: Yes, p.13-14 lines 362-370.

·         Author response 2: The study of Dellve and Erikkson was not included, as that study focused on the manager instead of the employee. In future research, it might be interesting to look at other levels of organizations. Changes to the manuscript: Yes, p. 13 line 345 and p.14 line 380-382.

Point 2: Your paper starts out by presenting the definition of sustainable employability by Van der Klink et al.. Their definition results from the capability approach by Sen which can be viewed a value-driven approach to work. In the discussion part of your paper, this approach is not much touched upon. Could you elaborate more on this lense and reflect on the results of the study in light of this lense?

·         Author response 2: It is a good suggestion to elaborate more on the value driven approach in the discussion part of the paper. We added the following text: “In line with the value-driven approach of Sen, ‘valuable work’ appears to be addressed effectively in three intervention studies. It seems that SE interventions including ‘valuable work’ enable a valuable work life and are as such appreciated by employees. All ‘valuable work’ outcomes are related to the individual level. However, it might well be that the work context rather than the individual facilitated these outcomes via the ‘valuable work’ component in the intervention. It is important to include a ‘valuable work’ component in SE interventions.” Changes to the manuscript: Yes, p.12 line 288-294.

Point 3: Your key words of searching for research that could fit your goal did not include sustainability, or sustainable HRM. I figure that you might have looked at all titles including 'sustain#' but this is not mentioned. Could you have missed some reference (also in view of the fact that you only found 7 and that these were primarily Dutch')?

·         Author response 3: We are constantly aware that our search might have been too narrow or the manual selection could have limited the number of hits unnecessarily. We therefore tried different search strategies and combinations of keywords. During the searches, many studies from outside the research field showed up (particularly studies focusing on sustainability in terms of improvement of the planet/environment). Therefore, we ultimately explicitly decided to focus on the combination of sustainability/sustainable AND work/employment. In the section of strengths and limitations of the Discussion, we added this extra explanation. Changes to the manuscript: Yes, p. 13 line 339- 343.

Point 4: Why did you only focus on quantitative studies?

·         Author response 4: As reported to the editor, we only focused on quantitative studies, because we were interested to examine, in quantitative/numerical terms, the effectiveness of SE interventions and their effective components. However, information on the process evaluation (if available) was used to figure out the underlying cause of the (in) effectiveness. In the method section, we accordingly explain why we only focused on quantitative studies. Changes to the manuscript: Yes, p. 2 lines 79-83.

·         Related to the previous bullet point about the process evaluations , much of the report in the Discussion when reflect on the limited effectiveness is based on the (qualitative) information from process evaluations, including information on relevant contextual factors. Changes to the manuscript: Yes, p. 12 line 268-322.

Point 5: I was wondering whether you can explain the 'operationalisation' of your key concepts. I know they are derived from the definion by Van der Klink et al. but I feel that some indicators are not in the right box. For example, quality of working life could also relate to valuable work. The right competences could also be in the box of productivity. Moreover, in some writings, vitality is a separate dimension of sustainable work, reflecting work engagement and motivation.  So, is this the right conceptualisation and the right operationalisation of the four core concepts? May be you could elaborate on this.

·         Author response 5: We agree that there is no consensus yet on how to operationalise SE core components. We have chosen four SE core components, which seems to be most in line with the definition of sustainable employability by van der Klink et al. There is no consensus yet among scholars on how the operationalize SE however. We incorporated this short explanation in the section of strengths and limitation in the discussion. Changes to the manuscript: Yes, p.13 line 354-357.

Point 6: Table 3 covers several pages and reading the tables is difficult when the heading of the column is presented some pages before. I wonder whether it would be better to present the 7 studies each in a separate table, since it is hard to have an overview.

Moreover, in the table it says training session, but the context information is missing. So may be the content of the intervention could be discussed in more detail, either in the text or in the table.

·         Author response 6: Indeed Table 3 covers several pages, but splitting the table into 7 tables may create too many tables. However, we repeated the heading of the columns on every page of Table 3. Furthermore, to get a good overview, we added in the Results section an extra paragraph with a detailed overview of the content of the seven interventions and their results.  Changes to the manuscript: Yes, p. 10 line 162- 218 (result section) and Table 3.

Point 7: p. 10, line 166: long-term perspective seems to be equalled to including all ages. I am not sure whether that makes sense.

·         Author response 7: We operationalized “long-term perspective” as SE interventions that focus on the future employability of employees and not only on the current employability. That implies an intervention focusing on all age groups rather than only older employees. Moreover, from a research perspective, we recommended that SE intervention studies should have a longer follow up period, to investigate the long-term effects as well. We introduced the operationalization of “long-term perspective” more clearly in the Introduction Changes to the manuscript: Yes, p.1 line 44-45, and p. 11 line 244.

Point 8: p. 10, line 182 and beyond. I would like to know why you think that the intervention can have negative effects. There is little explanation on the outcomes of the interventions. I think that this is valuable information that should be provided per study and with 7 that is doable. What is the underlying mechanism that is presented in each of the 7 studies or the context factors that explain the outcomes of the intervention. That would be very helpful for scholars and practitioners who want to learn from these studies and from your overview and evaluation. I think some thicker information would be of added value in that regard.

·         Author response 8: We agree with the reviewer to provide more explanation on the outcomes of the interventions and possible contextual factors as this indeed could be of added value for scholars and practitioners. Regarding the question why we think that interventions can have a negative effect, we described in line 277, a general explanation of negative effects in interventions, one that puts forward response shift as an explanation. Furthermore, in the same paragraph, we elaborate further on the effects of the interventions and the underlying reasons. Changes to the manuscript: Yes, p.12, line 277-306.

Point 9: p. 11, line 210: Are the contextual factors that might have disturbed the intervention mentioned in the results section. I might have missed these, but I think the outcomes of your analysis should be presented in the results section.

·         Author response 9: We provided information on the contextual factors in the Discussion section, where we discuss the possible underlying causes/reasons for the limited effectiveness. To better understand the (in)effectiveness of the interventions, several contextual factors might be relevant. Such information was obtained from the related process evaluations. In reviews of intervention effectiveness, information from process evaluations, we thought, is commonly used for reflect on the (in)effectiveness of the interventions (barriers, facilitators, implementation issues, contextual factors) rather than it being presented as a finding. We still have extended the Discussion on this issue. Changes to manuscript: p.12 line 277-322.

Point 10: p. 12. At page 12 you raise a lot of interesting and important issues. However, you do not provide an answer or some hints regarding the direction that one could take. For example, what do researchers need to do when workers have temporary jobs, are freelancing et cetera when it comes to longitudinal intervention research?

·         Author response 10: This a good suggestion of the reviewer. We added the following sentence to the Discussion: Researchers might consider the use of online surveys via national tax or social security registers to perform longitudinal intervention research. Changes to the manuscript: Yes, p. 14 line 406-407.

Point 11: p. 13 the references miss important information. Please check the reference list.

·         Author response 11: Thank you for this information. We revised the lay-out of the references list and provided additional information when possible for book references (page numbers, publisher etc.) Changes to the manuscript: Yes, p.15 reference list.